# The Serine Protease Autotransporters TagB, TagC, and Sha from Extraintestinal Pathogenic *Escherichia coli* Are Internalized by Human Bladder Epithelial Cells and Cause Actin Cytoskeletal Disruption

**DOI:** 10.3390/ijms21093047

**Published:** 2020-04-26

**Authors:** Pravil Pokharel, Juan Manuel Díaz, Hicham Bessaiah, Sébastien Houle, Alma Lilián Guerrero-Barrera, Charles M. Dozois

**Affiliations:** 1Institut national de recherche scientifique (INRS)-Centre Armand-Frappier Santé Biotechnologie, Laval, QC H7V 1B7, Canada; 2Department of Veterinary Medicine, Centre de recherche en infectiologie porcine et avicole (CRIPA), Faculty of Veterinary Medicine, Saint-Hyacinthe, QC J2S 2M2, Canada; 3Laboratorio de Biología Celular y Tisular, Departamento de Morfología, Universidad Autónoma de Aguascalientes (UAA), Aguascalientes 20131, Mexico; 4Department of Biology, Institut Pasteur International Network, Laval, QC H7V 1B7, Canada

**Keywords:** SPATEs, UTIs, cytotoxicity, serine proteases, 5637 bladder cells, mucin, gelatin, actin

## Abstract

TagB, TagC (*t*andem *a*utotransporter *g*enes *B* and *C*), and Sha (*S*erine-protease *h*emagglutinin *a*utotransporter) are recently described members of the SPATE (serine protease autotransporters of *Enterobacteriaceae*) family. These SPATEs can cause cytopathic effects on bladder cells and contribute to urinary tract infection in a mouse model. Bladder epithelial cells form an important barrier in the urinary tract. Some SPATEs produced by pathogenic *E. coli* are known to breach the bladder epithelium. The capacity of these newly described SPATEs to alter bladder epithelial cells and the role of the serine protease active site were investigated. All three SPATE proteins were internalized by bladder epithelial cells and altered the distribution of actin cytoskeleton. Sha and TagC were also shown to degrade mucin and gelatin respectively. Inactivation of the serine catalytic site in each of these SPATEs did not affect secretion of the SPATEs from bacterial cells, but abrogated entry into epithelial cells, cytotoxicity, and proteolytic activity. Thus, our results show that the serine catalytic triad of these proteins is required for internalization in host cells, actin disruption, and degradation of host substrates such as mucin and gelatin.

## 1. Introduction

Urinary tract infections (UTIs) present a broad range of symptoms and include urosepsis, pyelonephritis (or upper UTI, with infection in the kidney), and cystitis (or lower UTI, with bacteria infecting the bladder) [1,2]. Uropathogenic *Escherichia coli* (UPEC) is the main cause of community-acquired UTIs (about 80–90%) [3], and the ability of UPEC to establish a UTI is due to the expression of a variety of virulence factors. These factors include type 1 and P fimbriae (pili), flagella, capsular polysaccharides, iron acquisition systems, and toxins including hemolysin, cytotoxic necrotizing factor (CNF), and serine protease autotransporters of *Enterobacteriaceae* (SPATEs) [4]. 

The bladder urothelium constitutes a physical barrier to ascending urinary tract infections [5]. UPEC can produce toxins that damage bladder tissue and can lead to release of host nutrients and counter host defenses and innate immunity. A pore-forming toxin HlyA, can lyse erythrocytes and nucleated host cells [6], induce apoptosis [7], promote exfoliation of bladder epithelial cells and cause extensive uroepithelial damage [8,9,10,11]. Another UPEC toxin, cytotoxic necrotizing factor 1 (CNF1), has been reported to mediate bacterial entry into host epithelial cells [12], induce apoptotic death of bladder epithelial cells [13], and potentially promote bladder cell exfoliation [13]. SPATEs such as Sat, Pic, and Vat were also shown to affect bladder or kidney epithelial cells [14,15,16].

An important step to understand the role of SPATEs in UPEC pathogenesis is to elucidate molecular mechanisms underlying their effect on the bladder epithelium and during urinary tract colonization. The proteolytic activity of SPATEs is mediated by a serine protease catalytic triad of aspartic acid (D), serine (S), and histidine (H), wherein serine is the nucleophile, and aspartic acid interacts with histidine [17]. Mutations within the catalytic triad have been shown to abolish proteolytic activity in a number of SPATEs [15,17,18,19].

Recently, members of our group identified three new SPATEs: TagB, TagC (*t*andem *a*utotransporter *g*enes *B* and *C*), and Sha (*S*erine-protease *h*emagglutinin *a*utotransporter) in some strains of extra-intestinal pathogenic *E. coli* (ExPEC). In ExPEC strain QT598, *tagB* and *tagC* are tandemly encoded on a genomic island, and were present in 10% of UTI isolates and 4.7% of avian pathogenic *E. coli* (APEC) that we screened [20]. Further, Sha, which is encoded on a virulence plasmid in strain QT598 was present in 1% of UTI isolates and 20% of avian pathogenic *E. coli* [20]. The *tagBC* genes are also present in the genomes of sequenced UPEC strains such as multidrug-resistant CTX-M-15-producing ST131 isolate *E. coli* JJ1886 (Accession number CP006784), *E. coli* CI5 (Accession number CP011018), and multidrug-resistant uropathogenic *E. coli* strain NA114 (Accession number CP002797.2). When cloned into *E. coli* K-12, TagB, TagC, and Sha mediated autoaggregation, hemagglutination, and adherence to human HEK 293 renal and 5637 bladder cell lines, but did not contribute significantly to biofilm production [20]. Further, TagB and TagC exhibited cytopathic effects on the bladder epithelial cell line [20]. Following transurethral infection of CBA/J mice with a *tagBC* mutant or *sha* mutant, no significant difference in colonization was observed. However, the competitive fitness of a mutant derivative lacking all of the SPATEs present in QT598 was significantly lower in the kidney [20].

The purpose of this report was to more fully investigate the effects of the TagB, TagC, and Sha SPATEs on the 5637 bladder epithelial cell line focusing on the actin cytoskeleton. We also investigated potential entry of SPATE proteins within these bladder epithelial cells and whether they demonstrate mucinase or gelatinase activity.

## 2. Results

### 2.1. Processing and Secretion of TagB, TagC, and Sha Is Independent of the Serine Protease Motif

To evaluate the importance of the serine protease motif for processing and secretion of three novel SPATEs, we generated variant proteins of TagB, TagC, and Sha lacking the serine catalytic site. Plasmids expressing TagB, TagC, or Sha [20] were used as the templates for construction of site-directed mutant clones where the serine site was substituted for an alanine at residue S255, S252, and S258 respectively (Appendix A). Each of these three plasmids expressing mutant SPATEs, produced a high-molecular-weight protein (>100 kDa) in culture supernatants that corresponded to the expected size of the native protein, and also lacked breakdown products that are present in samples containing native SPATEs that exhibit some autoproteolytic activity (Figure 1A, asterisks). This demonstrated that the serine protease motif is not necessary for SPATE secretion and release from bacterial cells. 

To further localize each of the SPATEs expressed from plasmids in *E. coli* BL21 on the bacterial cell surface, we used immunogold labeling and transmission electron microscopy. Polyclonal antibodies against the entire secreted Vat SPATE [20] were used as they were shown to strongly cross-react and recognize conserved epitopes of the other SPATEs. Thus, we used these polyclonal “SPATE antibodies” for the detection of other SPATEs in our experiments. 

*E. coli* BL21 pBCsk+ expressing TagB, TagC, and Sha were immunogold-labeled (Figure 1B–D); demonstrating localization of these SPATEs on the bacterial surface, as well as release into culture supernatant. The inset depicts the heavily concentrated proteins on the bacterial surface that appears to cluster around each other and produce fiber-like aggregates on the cell surface (white chevron, Figure 1B–D). *E. coli* BL21 bacteria containing only the empty plasmid vector were not labeled after immunogold labeling with anti-SPATE antibodies and secondary anti-rabbit immunoglobulin conjugated to 10-nm gold particles (Figure 1E). 

### 2.2. The Serine Catalytic Motif of SPATEs Is Not Required for Autoaggregation or Hemagglutination Activity 

To determine if the serine catalytic site of each of the three SPATEs is involved in autoaggregation or hemagglutination activity, we tested these phenotypes with our mutant SPATEs as described in [20]. Macroscopic analysis of autoaggregation of *E. coli* BL21 expressing TagB S255A, TagC S252A, or Sha S258A showed that bacterial cells settled at the bottom of the tube under static incubation similar to their respective native protein-expressing clones. The percentage of reduction in turbidity is given as a percentage of the initial OD_600_ value and was similar for both mutant and native proteins (Figure 2). The reduction in turbidity of the negative control *E. coli* BL21 pBCsk+ was significantly lower compared to clones expressing TagB, TagC, Sha, or AIDA-1 (positive control) (Figure 2). AIDA-1 (Adhesin Involved in Diffuse Adherence) of *E. coli* is a characterized self-associating autotransporter protein which mediates bacterial cell–cell interactions and autoaggregation [21].

These results show that inactivation of the serine catalytic site in each SPATE does not affect autoaggregation. It is therefore likely that other motifs or residues present in these proteins contribute to autoaggregation of bacterial cells. 

Likewise, Sha S258A showed similar hemagglutination of human blood as reported previously for the Sha native protein [20]. When cloned in the hemagglutination negative *E. coli* strain ORN172, there was no hemagglutination activity for either the native or mutant TagB or TagC proteins. In addition to autoaggregation and hemagglutination activity, the adherence capability of the serine catalytic site mutants of the SPATEs to 5637 human bladder epithelial cells was not affected (Appendix A). Hence, loss of the serine catalytic site did not affect autoaggregation, adherence, or hemagglutination phenotypes associated with each of the SPATEs compared to the native proteins.

### 2.3. Cytopathic Effect of TagB and TagC Requires the Serine Protease Motif

To assess the role of the serine protease motif for the cytopathic effect of SPATEs, extracts of supernatants of the different SPATEs (30 μg of protein per well) were incubated with human bladder epithelial cell line 5637 for 5h. Then the cells were fixed, stained with Giemsa stain, and observed by light microscopy. Cytopathic changes (dissolution in cytoplasm, enlargement of the nucleus with vacuoles) observed under the microscope for TagB and TagC (Figure 3A) were absent from cells treated with either TagB or TagC proteins lacking the serine protease active site. In addition, no significant morphological changes were observed with cells treated with either Sha or the mutant, Sha S258A, protein. No cytopathic effect was observed after treatment of cells with concentrated filtered supernatant from *E. coli* BL21 pBCsk+ (empty vector) or media alone (Figure 3A). To examine this cytopathic effect quantitatively, we measured lactate dehydogenase (LDH) release from epithelial cells incubated with each of SPATEs or their respective catalytic site mutant proteins. There was release of LDH after 5 h upon exposure of cells to TagB or TagC. However, the catalytic site mutant proteins did not release LDH from cells (Figure 3B). Further, no LDH release was detected from cells treated with either Sha or its catalytic site mutant variant (Figure 3B), indicating that cytotoxicity to human bladder epithelial cells by TagB and TagC was dependent on the serine protease catalytic site.

### 2.4. Exposure to TagB, TagC, or Sha Alters Actin Distribution in Bladder Epithelial Cells

Based on the cellular changes seen with bladder epithelial cells after exposure to TagB and TagC, we hypothesized that TagB and TagC could alter the distribution of cytoskeletal components such as actin, with actin being one of the most abundant intracellular proteins in the eukaryotic cell. So, to examine the effect on F-actin cytoskeleton organization, 5637 bladder cells were incubated with native and mutant TagB, TagC, or Sha (30 μg of protein per well) for 5 h at 37 °C, stained with fluorescently labeled phalloidin, and then observed under confocal microscopy. Cells treated with the supernatant extract from the empty vector containing clone were uniform, smooth-edged, and contained clearly visible actin stress fibers (yellow triangle) and strong actin staining around the cell (Figure 4A). By contrast, bladder cells treated with TagB showed reduced actin stress fibers and less actin staining (Figure 4A). Bladder cells treated with TagC, also had a pronounced effect on the cytoskeleton as demonstrated by the absence of actin stress fibers and reduced levels of actin staining. Sha treated cells showed a loss of actin stress fibers and the presence of punctate patterns of actin within the cytoplasm of the cells (yellow arrowheads, Figure 4A). By contrast, the TagB, TagC, and Sha mutants lacking the serine protease catalytic sites demonstrated no changes in the actin cytoskeleton and had actin stress fibers similar to negative control cells, indicating that the serine protease activity of these SPATEs mediates the changes in actin distribution within bladder cells. To quantify the level of phalloidin binding, we measured the staining intensity and distribution of fluorescence of phalloidin around each cell using ImageJ software [22]. Fluorescence intensity for cells was calculated using the channel for actin staining. In comparison with the negative control (empty vector), the density of F-actin staining was significantly lower in cells treated with TagB, TagC, or Sha. Cells treated with the serine catalytic site mutant proteins, demonstrated F-actin staining that was greater when compared to cells treated with the native SPATE proteins (Figure 4B). Overall, these results demonstrate that these SPATEs alter the cytoskeleton and reduce the distribution of actin in bladder epithelial cells.

### 2.5. SPATE Entry into Bladder Epithelial Cells Is Dependent on the Serine Protease Active Site

We previously showed that TagB and TagC demonstrated cytotoxicity as measured by lactate dehydrogenase (LDH) release from epithelial cells within 5 h [20]. This toxicity could be due to the interaction of the SPATEs with targets inside host cells. So, to gain insight into the potential internalization of these SPATEs, we employed immunofluorescence labeling of proteins followed by visualization using confocal or immunogold electron microscopy. Firstly, confocal Z-sections (optical slices) of 5637 bladder cells treated with SPATEs were examined to determine if SPATEs were translocated within cells. After 5 h of incubation, TagB, TagC, and Sha (red color) were found within cells as evidenced by cell sectioning analysis (Figure 5A). By contrast, the serine active-site mutant variants were unable to enter epithelial cells and were not detected (absence of red staining) (Figure 5A), suggesting that serine protease activity is needed for the entry of SPATEs within cells. Interestingly, TagB within cells also co-localized with actin (green color) in the outer border of the cell (Figure 5A). Further, cells incubated with serine mutant variants of SPATEs did not enter cells, and these cells also produced actin stress fibers (Figure 5).

Analysis of thin-sections of SPATE-treated cells using immunogold staining and transmission electron microscopy (TEM) also confirmed the intracellular localization of all three SPATEs within cells. TagB and Sha were found in the cytoplasm, whereas TagC was present in the nucleus (Figure 6). However, in multiple independent experiments, we failed to detect the presence of serine mutant variants of TagB, TagC, or Sha within cells. The serine catalytic-site mutant proteins when visualized were almost exclusively observed on the extracellular surface of cells as seen in cells treated with TagB S255A (Figure 6D).

### 2.6. Sha Exhibits Serine Protease-Dependent Mucinase Activity

Epithelial cell damage caused by SPATEs was shown to require protease activity, and some other SPATEs were previously shown to demonstrate activity against host proteins such as mucin or gelatin [18,23]. Further, we also tested for mucinase activity, since two of the novel SPATEs identified in APEC QT598, Sha and TagB [20], belong to the class 2 SPATE family whose members have been shown to demonstrate mucinolytic activity. Clones of *E. coli* BL21 expressing each of the SPATEs were grown on agar plates containing 0.5% porcine gastric mucin for 24 h at 37 °C, followed by amido black-staining. Plates containing clones growing on discs expressing Sha revealed clear zones of mucin lysis (Figure 7A) and the lysis zone produced by Sha was intermediate when compared to clones expressing either Tsh (positive control) or Vat. Mucin containing plates had a clearing zone with a diameter of 3.9 ± 0.1 cm after exposure to Sha expressing bacteria, which was less than following exposure to Tsh expressing cells (4.2 ± 0.1 cm), but more than following exposure to Vat expressing cells (3.7 ± 0.2 cm). By contrast, TagB and TagC were mucinase-negative as evidenced by the absence of any clearing zones (Figure 7B). Further, the critical role of the serine catalytic site of Sha for mucinase activity was demonstrated with the clone expressing Sha S258A, which did not produce a zone of mucin lysis (Figure 7B). The clone containing only the empty vector (negative control) did not grow well in the presence of mucin and also demonstrated no clearing zone. When mucin was treated with culture filtrates of SPATE proteins (Figure 7C), it was not degraded by either TagB, TagC, or in the negative control (empty vector). Sha as well as Tsh and Vat degraded mucin, whereas the serine protease mutant of Sha, Sha S258A, did not. Hence, the serine catalytic site of Sha is required for mucinase activity.

### 2.7. TagC Exhibits Serine Protease-Dependent Gelatinase Activity

Some SPATEs were previously reported to degrade extracellular matrix proteins such as collagen and gelatin [23]. We previously demonstrated that TagB, TagC, and Sha could mediate increased adherence to chicken fibroblasts [20], which are cells that are associated with connective tissues and produce extracellular matrix proteins such as collagen. The hydrolyzed form of collagen—gelatin was used as a substrate to test for potential gelatinase activity from supernatant extracts containing SPATEs. Culture supernatant filtrate from *Pseudomonas aeruginosa* was used as a positive control, since it is known to demonstrate gelatinase activity [24]. Samples were incubated with 1% bovine gelatin for 48 h at 37 °C. Culture filtrates containing TagC as well as other SPATEs EspC, Tsh, and Vat demonstrated gelatinase activity (Figure 8A). By contrast, neither TagB nor Sha demonstrated gelatinase activity, since high-molecular-weight bands, indicating intact gelatin, remained after exposure to these SPATEs. Further, gelatinase activity from TagC was shown to be dependent on the serine protease motif, since the *E. coli* clone expressing a serine active site mutant protein, TagC S252A, did not generate a hydrolysis zone on medium containing 1% gelatin, whereas the TagC expressing clone did exhibit a hydrolysis zone (Figure 8B).

## 3. Discussion

Colonization of the bladder is vital for UTI pathogenesis and UPEC deploys an array of virulence factors to infect and colonize the bladder, including secreted toxins [25]. Hemolysin A [8,9], UpxA (TosA) [26], cytotoxic necrotizing factor-1 (CNF-1) [27,28], and a variety of SPATEs (serine-protease autotransporters of *Enterobacteriaceae*) [29] are known toxins of host cells that are produced by some UPEC strains. The recent identification of new members of the SPATEs family present in some pathogenic *E. coli* and their cytotoxic activity on bladder cell lines [20], led us to further investigate mechanisms underlying the cytotoxic and proteolytic activity of the TagB, TagC, and Sha SPATEs on an established human urinary bladder cell line [30,31] and other properties of these virulence-associated proteins.

TagB, TagC, and Sha proteins demonstrated autoaggregating activity, and also promoted adherence of *E. coli* strain BL21 to the human HEK 293 renal and 5637 bladder human cell lines. Further, Sha also contributed to increased biofilm production [20]. SPATEs present on the bacterial surface are likely to contribute to the autoaggregation phenomenon. (Figure 1A). TagB and TagC also exhibited cytopathic effects on the bladder epithelial cell line. Further, we also previously determined that proteolytic activity of these SPATEs was strongly inhibited upon addition of serine protease inhibitor (PMSF), providing evidence for the importance of the serine protease motif in the activity of these SPATEs [20]. To further investigate the role of the serine protease activity, we generated catalytic site mutants of these three SPATEs. It is of note that the serine protease consensus motif (GDSGS) is conserved among different members of SPATEs [20,32,33,34,35]. Importantly, loss of the serine active site did not affect the processing or secretion of the SPATE proteins into the extracellular milieu (Figure 1A). Further, loss of the serine active site also eliminated any autoproteolytic activity (Figure 1A). Similarly, autoproteolytic activity has also been reported for other SPATEs including EspP, Sat, Pic [36], and for AspA autotransporter from *Neisseria meningitidis* [37]. Thus, from our results, it is clear that the processing of the passenger domain across the bacterial surface and autocatalytic activities of the TagB, TagC, and Sha is independent of the proteolytic serine site.

We investigated the role of the serine catalytic site of TagB, TagC, and Sha in autoaggregation or hemagglutination, as either SPATE protease activity on the bacterial or host cell surfaces could have possibly mediated these phenotypes. For instance, cleavage could have led to certain domains within the protein, leading to exposure of hydrophobic sites which could promote aggregation [38]. However, the serine protease site was not required for TagB, TagC, or Sha-mediated aggregation (Figure 2). These results indicate that other specific SPATE structural domains are likely to be responsible for aggregation. However, importantly, the autoaggregation phenotype is not a generalized phenotype of SPATEs, since in previous experiments, both the Tsh and Vat SPATEs did not demonstrate any autoaggregation phenotype [20]. Currently, the molecular mechanism of autoaggregation of TagB, TagC, and Sha is unknown. Unlike the three SPATEs described herein, loss of the active site serine of the Hap adhesin, a *Haemophilus influenzae* serine protease autotransporter, abrogated autoproteolytic processing leading to retention of this AT protein on the bacterial cell surface [39]. In fact, the increase in Hap present on the bacterial surface also increased aggregation, formation of microcolonies, and adherence of *H. influenzae* to host cells [40]. With regards to hemagglutination activity of the Sha protein, the serine active site was also dispensable. We found that Sha S258A hemagglutinated human blood with a similar titer to the native Sha protein. Similarly, a Tsh S259A variant protein was also able to bind to avian erythrocytes, turkey hemoglobin, collagen IV, fibronectin, and laminin [41]. Considering that the TagB S255A, TagC S252A, and Sha S258A variant SPATEs all retained the respective phenotypes present in the native SPATE proteins, this suggests that, despite lacking catalytic activity, that these variants are likely to have maintained a properly folded conformation.

In contrast to adherence or aggregation phenotypes, the presence of a serine protease motif was clearly required for cytotoxicity and entry of the SPATE proteins into bladder epithelial cells. In this study, the TagB S255A and TagC S252A mutant proteins were no longer cytopathic. Our results are similar to those described for other SPATEs [15,41,42] which have demonstrated a key role for the serine active site with regards to any native proteolytic or cytopathic activity of SPATEs on protein substrates or host cells.

Since TagC shares 60% identity/74% similarity with another SPATE, EspC, a non-LEE-encoded enterotoxin of enteropathogenic *E. coli* (EPEC) which causes cytotoxic effects and cleavage of cytoskeletal actin-associated protein [43]; we explored potential cellular targets in relation to the cytopathic effect observed in bladder epithelial cells. Following treatment with either TagB, TagC, or Sha, reorganization of the cytoskeleton and loss of actin stress fibers were seen in bladder epithelial cells (Figure 4). The effect of TagC was severe with faint staining remaining for actin compared to TagB interaction with cells. Exposure to Sha caused punctate localization of actin within the cytoplasm. Diminished actin staining and the formation of punctate actin accumulation suggests that each of these SPATEs are targeting the actin cytoskeleton or other cellular targets that lead to modifications in actin fiber formation or distribution within bladder cells. As expected, alterations in actin distribution were absent from bladder cells exposed to the serine catalytic site mutant variant proteins TagB S255A, TagC S252A, or Sha S258A, confirming the critical cytopathic role of serine protease activity.

Many pathogens exploit host actin for various stages of infection, including cellular invasion, intracellular replication, and dissemination by different mechanisms [44,45]. Specifically, during UTIs, UPEC utilizes the Rho family GTPase member Rac1 to mediate actin polymerization for *E. coli* bladder epithelial cell invasion [46]. It has been well documented that there is a relation between intracellular growth of UPEC in the bladder epithelium and the host F-actin cytoskeleton [47]. Based on the observation of actin rearrangement observed in bladder cells, it is also possible that the TagB, TagC, and Sha SPATEs might also contribute to UPEC invasion of the bladder epithelium, as these proteases may promote adhesion and loss of integrity of the protective epithelial barrier which could increase bacterial entry into epithelial cells as well as increase entry and systemic spread of the bacteria to other tissue sites during infection.

Before reaching the epithelial cell surface in the urinary tract, bacteria must cross the protective mucus layer that is coated with mucin [48]. Mucin serves as a primary antibacterial defense in the bladder and contributes to host innate defense by providing a barrier and by trapping bacteria [49]. Many pathogens can invade or reduce the viscosity of mucin by cleaving it [50,51,52]. Certain SPATEs, belonging to the Class 2 family, including Pic [18,53], PicC of *Citrobacter rodentium* [53], and Tsh, demonstrate mucinase activity [54]. We, therefore, tested whether any of the three novel SPATEs were mucinolytic, and only Sha was identified as a mucinase (Figure 7). The zone of mucinolytic activity of Sha was intermediate when compared to Vat and Tsh and, as has been shown for Pic [18], the serine catalytic site of Sha was required for mucinase activity. From this standpoint, it is interesting to note that in strain QT598, 3 of the 5 SPATEs (Tsh, Vat, and Sha) demonstrate mucinase activity [20] which might facilitate bacterial colonization by degrading mucus to overcome the mucous barrier at the interface of epithelial surfaces. TagC was also shown to degrade gelatin, which is the hydrolyzed form of collagen, although this activity was absent from Sha and TagB. Collagen is an abundant and ubiquitous extracellular matrix protein that forms an essential component of connective tissues [55]. From this standpoint, the TagC protease may contribute to tissue invasion and systemic spread of ExPEC by degradation of extracellular matrix proteins. As expected, the activity of TagC on gelatin was also dependent on the active serine catalytic site. Similarly, Pic [23] also demonstrated gelatinase activity that required an active serine catalytic site.

Previous reports have described different mechanisms of internalization of SPATEs and types of cytoskeletal damage in various epithelial cells in vitro. The Pet SPATE from enteroaggregative *E. coli* (EAEC) is internalized by a retrograde trafficking pathway [56] through the Pet host cell receptor, cytokeratin 8 [57]. Once internalized, Pet causes loss of actin stress fibers due to the breakdown of spectrin [58,59]. Internalization of EspC by EPEC requires the type 3 secretion system [60] and leads to cleavage of cytoskeletal proteins [43]. Sat is secreted by UPEC, enters the cell by an unknown mechanism, and localizes to the cytoskeletal fraction of fodrin/spectrin and integrin present within bladder and kidney epithelial cells [15]. In the present report, we have demonstrated that TagB, TagC, and Sha are also internalized in bladder epithelial cells by a mechanism that requires an active serine catalytic domain. We used confocal Z-sections to verify the intracellular localization of the SPATEs within human bladder epithelial cells. Of note, we observed the internalization of TagB, TagC, and Sha within bladder cells after 5 h and this was concomitant with diminished fluorescence staining of actin in the vicinity of the localized SPATEs. This observation was pronounced following exposure to TagB, and TagB was shown to be closely associated with actin. Furthermore, to confirm the internalization of SPATEs within bladder epithelial cells, we carried out immunogold TEM of cross-sections of cells to demonstrate SPATE proteins within epithelial cells. TEM demonstrated localization of TagB and Sha in the cytoplasm, whereas TagC targeted the nucleus. We speculate that, since TagC has previously been shown to promote nuclear enlargement [20], TagC may alter nuclear targets and elicit a significant increase in nuclear size. The entry of these SPATEs into host bladder cells does not require a type 3 secretion mechanism since it is absent from *E. coli* QT598 and *E. coli* BL21, and SPATE proteins from bacterial supernatants entered bladder epithelial cells directly. Future studies will elucidate the cytoplasmic or nucleo-cytoplasmic shuttling pathways that mediate the entry and trafficking of these three SPATEs. Importantly, the serine catalytic site was required for cell entry and cytotoxicity of all three SPATEs, since serine protease active site mutants were unable to enter cells or cause any cytopathic effects, further demonstrating a critical role for the serine catalytic site of these SPATEs.

Taken together, the TagB, TagC, and Sha SPATE proteins mediate multiple activities. These include adhesion, aggregation, cytopathic effects, mucinase and gelatinase activities that may collectively contribute to different stages of bacterial infection including initial colonization, invasion of host epithelia, and an increased potential for systemic infection.

## 4. Materials and Methods

### 4.1. Ethics Statement

This study was performed in accordance with the ethical standards of the University of Quebec, INRS. A protocol for obtaining biological samples from human blood donors was reviewed and approved by the ethics committee—*Comité d’éthique en recherche* (CER 19-507, approved November 19, 2019) of INRS.

### 4.2. Bacterial Strains, Plasmids, and Growth Conditions

*E. coli* clones expressing TagB, TagC, or Sha were described previously [20]. All DNA constructs were transformed into *E. coli* strain BL21 or the type 1 fimbriae *fim*-negative *E. coli* strain ORN172. Strains were grown at 37 °C on solid or liquid Luria-Bertani medium (Alpha Bioscience, Baltimore, MD, USA) supplemented with the appropriate antibiotics when required at concentrations of 100 µg/mL ampicillin, 30 µg/mL chloramphenicol, or 50 µg/mL of kanamycin. Strains, plasmids, and primers are listed in Table 1.

### 4.3. Site-Directed Mutagenesis

Site-directed mutagenesis was performed using the Q5^®^ Site-Directed Mutagenesis kit as specified by the manufacturer. pIJ548, pIJ544, and pIJ553 were used as a template for the construction of the serine catalytic site mutants TagB S255A (pIJ554), TagC S252A (pIJ555), and Sha S258A (pIJ556) at 25 to 50 ng per reaction with 10 pmol of each of the complementary primers. Primers used to generate the single point mutation substituting alanine for serine for TagB were 5′-TCCCGGTGACGCCGGCTCTCCT-3′ and 5′-GTACCGTAGGTTGAGAGTG-3′; TagC were 5′-AGGAGGAGACGCCGGTTCCGGA-3′ and 5′-GTCACTTCATTATAAAATCCACC-3′; and Sha were 5′-GGCTGGTGATGCCGGTTCTCCGC-3′ and 5′-TCACCATAGATCGGTAATAC-3′. Following mutagenesis, all constructs were verified by sequencing at the proteomics platform of the Institut de Recherche en Immunologie et en Cancérologie (IRIC) of the Université de Montréal (Montréal, QC, Canada).

### 4.4. Recombinant Protein and Antibody Preparation

Expression and purification of SPATE proteins from concentrated filtered culture supernatant fractions were obtained as described previously [20] and the extract was checked by silver staining before each assay. Antibodies against ~ 112 kDa Vat protein were used to generate a Vat-specific rabbit polyclonal antibody, according to a standard protocol [64] (Laboratorio de Biología Celular y Tisular, Departamento de Morfología, Universidad Autónoma de Aguascalientes (UAA), Aguascalientes, Mexico). Since SPATE proteins contain some highly conserved epitopes, anti-Vat antibodies were used to detect and label each of the SPATE proteins. The alignment of the passenger domain of Vat with TagB, TagC, and Sha share identities of 39%, 30%, 56%, respectively. Specific epitopes are not established but Vat-antibodies demonstrate multiple conserved residues (Appendix A) and strong immune cross-reactivity. Cross-reactivity of antibodies raised against other SPATEs have also been reported. Antibodies raised against Pet protein (45% identity with EspC and 60 gaps) cross-reacted with EspC [65]. Likewise in the supernatant of CFT073, anti-Pic (44% identity with Vat and 76 gaps) antibodies were used to detect Vat and PicU SPATEs [66]. Polyclonal antisera adsorption was done by incubating the filtered supernatant of *E. coli* BL21 pBCsk+ without insert with a 1:50 dilution of the Vat polyclonal antiserum for 1 h at room temperature under mild agitation followed by centrifugation at 2000× *g* for 5 min at 4 °C.

### 4.5. Autoaggregation and Hemagglutination Tests

Autoaggregation of bacterial cells was measured by a settling assay as performed previously [20]. The sedimentation of 10 mL of each culture of *E. coli* BL21 cells expressing native or serine active site mutant SPATEs were adjusted to an OD_600nm_ 1.5 from an overnight culture grown at 37 °C in liquid Luria-Bertani medium. Then, they were monitored for a reduction in turbidity from the top of the tube which was left at 4 °C for 3 h. The reduction of turbidity was plotted as a ratio against the initial turbidity.

For hemagglutination assays, human blood cells (RBCs) were washed and resuspended in PBS at a final concentration of 3% using a protocol adapted from [67]. The *E. coli fim*-negative K-12 strain ORN172 expressing either native or serine active site mutant SPATEs was grown overnight at 37 °C in Luria-Bertani medium, harvested and adjusted to an optical density (O.D._600nm_) of 60. Suspensions were serially diluted in 96-well round-bottom plates containing 20 µL of PBS mixed with 20 µL of 3% red blood cells and incubated for 30 min at 4 °C.

### 4.6. Epithelial Cell Culture

The 5637 bladder epithelial cell line was routinely cultured in RPMI 1640 medium (Thermo Fisher Scientific) supplemented with 10% heat-inactivated FBS at 37 °C in humidified 5% CO_2_, and 2 × 10^5^ cells/well were seeded into eight-well chamber slides (Thermo Fisher Scientific, Waltham, MA, USA) and allowed to grow to 75% confluence.

To determine cytopathic effects on bladder cells, a final concentration of 30 μg/mL of native SPATEs or the serine catalytic site mutants were added directly to monolayers and incubated for 5 h in RPMI 1640 medium at 37 °C with 5% CO_2_. Cells were then washed twice with PBS (phosphate-buffered saline), fixed with 70% methanol, and stained with Giemsa stain. Cell morphology was analyzed at a magnification of ×20 with standard bright-field light microscopy. For the lactate dehydrogenase assay, supernatant from cells treated with native or mutant SPATEs were collected and the release of LDH in cell culture supernatants were quantified by using the CytoTox 96^®^ Non-Radioactive Cytotoxicity Assay kit (Promega, Madison, WI, USA). Maximum LDH release (positive control) was determined by adding lysis solution (provided in the kit) to the non-infected cells.

For fluorescence actin-staining and immunostaining assays, cells were fixed with 3.0%–4.0% formaldehyde in PBS, washed, permeabilized by addition of 0.1% Triton X-100-PBS, stained with 0.05 μg of Alexa Fluor 488-phalloidin/mL (AAT Bioquest, Sunnyvale, CA, USA) at 37 °C for 1 h and counterstained with ProLong Gold/DAPI antifade reagent (Invitrogen, Carlsbad, CA, USA). After image acquisition using confocal microscope, the actin staining intensity was quantified by measuring mean gray value (mean pixel intensity) in ImageJ (https://imagej.nih.gov/ij/) [22]. The cells of interest as well as background with no fluorescence were selected manually to analyze the areas integrated intensities and mean gray value. The value was then corrected and total fluorescence (CTF) was calculated as CTF = Integrated Density – (Area of selected cell X Mean fluorescence of background readings). The averaged corrected mean gray value was used to generate relative quantitative comparison of fluorescence intensity.

SPATE protein localization in bladder cells was detected by immunofluorescence. Treated cells were fixed, permeabilized, and incubated with blocking solution (PBS with 5% BSA) for 1 h at 37 °C. Samples were then incubated with rabbit anti-SPATE polyclonal antibodies (UAA, Mexico) for 2 h at 37 °C. This was followed by incubation with secondary antibody Alexa Fluor 594-labeled goat anti-rabbit IgG antibody (Thermo Fisher Scientific, Waltham, MA, USA). Samples were mounted and imaged with the 60X objective of an LSM780 confocal microscope (Carl Zeiss microscopy Gmbh, Jena, Germany). Images were processed with ZEN 2012 software (Carl Zeiss microscopy Gmbh, Jena, Germany).

### 4.7. Electron Microscopy

Immunogold labeling of bacteria was carried out by culturing *E. coli* BL21 expressing different SPATEs in Luria-Bertani medium supplemented with 30 µg/mL chloramphenicol for 5 h. Bacterial suspensions (50 µL) were spotted on nickel-coated TEM grids. After 15 min, liquid was wicked away with bibulous paper and blocked with drops of PBS containing 1% ovalbumin for 15 min. A blocking solution was exchanged with a drop of SPATE antiserum diluted 1:100 in PBS. After 15 min, excess fluid was wicked away with bibulous paper and exchanged for PBS containing 1% ovalbumin drops for 5 min. The wash was repeated and then incubated in suitable goat anti-rabbit IgG (H+L), Alexa Fluor 488–10 nm colloidal gold secondary antibodies (Thermo Fisher Scientific, Waltham, MA, USA) diluted 1:250 in incubation solution. After 15 min, grids were washed twice with PBS drops and rinsed twice with distilled water. Grids were dried with bibulous paper and imaged on a Philips CM-100 transmission electron microscope.

For immunogold labeling of epithelial cell thin sections, cells were fixed in 0.1% glutaraldehyde + 4% paraformaldehyde in cocodylate buffer at pH 7.2, and post-fixed in 1.3% osmium tetroxide in collidine buffer. After dehydration by successive passages through 25, 50, 75, and 95% solutions of acetone in water (for 15–30 min each) samples were immersed for 16–18 h in SPURR: acetone (1:1). Samples were then embedded in BEEM capsules using SPURR resin with the ELR-4221 kit (Polysciences Inc, Warrington, PA, USA) followed by placing the capsules at 60–65 °C for 20–30 h to polymerize the resin. After resin polymerization, samples were cut using an ultramicrotome (Ultratome) and were put onto Formvar and carbon covered-copper 200-mesh grids treated with sodium metaperiodate and were blocked with 1% BSA in PBS. Grids were then incubated with primary antibodies, washed, and incubated with goat anti-rabbit IgG (H+L), Alexa Fluor 488–10 nm colloidal gold secondary antibodies (Thermo Fisher Scientific, Waltham, MA, USA). After washing, samples were contrasted with uranyl acetate and lead citrate and subsequently visualized using a Philips EM 300 transmission electron microscope.

### 4.8. Cleavage of Protein Substrates

For mucinase activity, cultures of *E. coli* BL21 expressing SPATEs were incubated for 24 h at 37 °C on a medium containing 1.5% agarose and 0.5% porcine gastric mucin (Sigma-Aldrich, St. Louis, MI, USA). Plates were subsequently stained with 0.1% amido-black in 3.5 M acetic acid for 15 min, followed by destaining with 5% acetic acid and 0.5% glycerol for 6 h to overnight. Zones of mucin lysis were visualized as discolored halos around colonies. For the Periodic Acid Schiff (PAS) assay to detect mucin degradation [53], 5 µg of each SPATE protein were incubated with 5 µg of 0.5% porcine gastric mucin (Sigma-Aldrich) in 30 μL of MOPS buffer and incubated for 48 h at 37 °C. Treated samples were electrophoresed on an 8% SDS-PAGE gel and the gel staining was developed using a colorimetric Pierce™ Glycoprotein Staining kit (ThermoFisher Scientific, Waltham, MA, USA).

For gelatinase activity, 5 µg of each SPATE protein were incubated with 5 µg of bovine skin gelatin (Sigma-Aldrich, St. Louis, MI, USA) in 30 μL of MOPS buffer and incubated for 48 h at 37 °C. Samples were then boiled with Laemmli sample buffer, were electrophoresed on an 8% SDS-PAGE gel and then resolved by Coomassie blue staining. In addition, gelatinase activity was also tested by growing the clones on agar plates containing 1.5% agarose and 1% bovine skin gelatin for 48 h at 37 °C. Plates were subsequently stained with 0.1% amido-black in 3.5 M acetic acid for 15 min, followed by destaining with 5% acetic acid and 0.5% glycerol for 6 h to overnight. Zones of gelatin lysis consist of discolored halos around colonies.

### 4.9. Statistical Analysis

Experimental data were expressed as a mean ± standard error of the mean (SEM) in each group. The means of groups were combined and analyzed by a two-tailed Student *t*-test for pairwise comparisons and analysis of variance (ANOVA) to compare means of more than two populations. A *p* value of <0.05 was considered statistically significant. All data were analyzed with the Graph Pad Prism 7 software (GraphPad Software, San Diego, CA, USA).

## 5. Conclusions

In conclusion, TagB, TagC, and Sha are novel SPATEs that demonstrate different proteolytic activities on different substrates as well as distinct cytopathic effects on bladder epithelial cells. Additional molecular *in vitro* and *in vivo* studies are in progress in an effort to understand the link between protease activity of the TagB, TagC, and Sha SPATEs and how these proteases disrupt or alter the actin cytoskeleton during ExPEC infections. It will be of further interest to also investigate their potential interactions with other host cells or extracellular matrix proteins, and determine how these relatively large proteins (generally greater than 100 kDa) manage to enter host cells through serine protease activity and what specific trafficking pathways may be involved in their localization or association with specific cellular compartments.

## Figures and Tables

**Figure 1 ijms-21-03047-f001:**
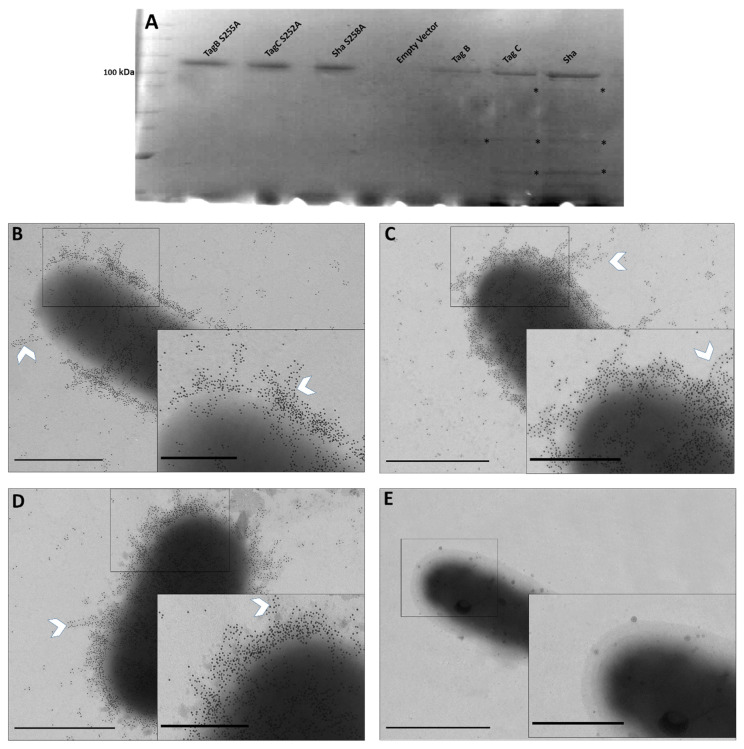
(**A**) Silver stained SDS-PAGE analysis of concentrated supernatants of *E. coli* BL21 expressing SPATE proteins. Filtered supernatants from clones expressing TagB, TagC, and Sha or the variant TagB S255A, TagC S252A, and Sha S258A proteins were concentrated through Amicon filters with a 50 kDa cutoff. Samples containing 5 µg of protein were migrated and stained with silver stain. (**B**–**E**) Immunogold Electron Microscopy (EM) of SPATEs (Serine protease autotransporters of *Enterobacteriaceae*) localized to the outer membrane and extracellular medium. Immunogold-TEM micrographs of SPATEs using SPATE-specific antiserum. Bacteria were cultured to 0.6 OD_600nm_ in Luria-Bertani medium. *E. coli* BL21 pBCsk+ expressing TagB (**B**), TagC (**C**), and Sha (**D**) labelled with immunogold particles. (**E**) *E. coli* BL21 pBcsk+ (vector only control) shows no immunogold staining. Insets represents boxed areas of higher magnification showing clustering of SPATE proteins. All images were acquired at ×17,000 magnification; scale bars represent 1 µm, and 0.5 µm (Insets).

**Figure 2 ijms-21-03047-f002:**
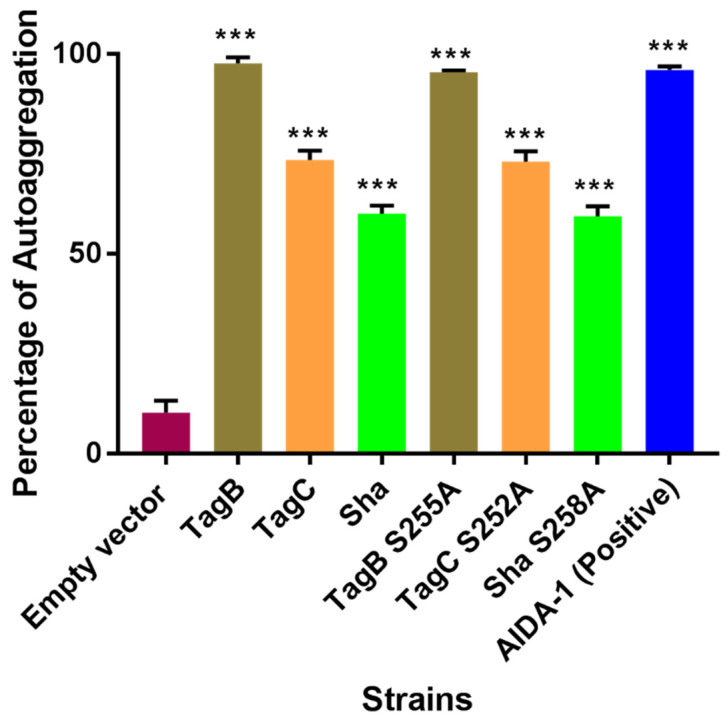
The autoaggregation phenotype is independent of the serine protease motif. Clones of *E. coli* BL21 expressing TagB, TagC, Sha, or their respective serine-site mutants were grown 18 h and adjusted to an OD_600_ of 1.5 and left to rest at 4 °C. Samples were taken at 1 cm from the top surface of the cultures after 3 h to determine the change in OD_600_. Assays were performed in triplicate, and the rate of autoaggregation was determined by the mean decrease in OD_600nm_ after 3 h. *E. coli* BL21 pBCsk+ vector without insert (empty vector) was used as a negative control and the AIDA-1 autotransporter was the positive control for autoaggregation. Error bars represent standard errors of the means (*** *p* < 0.001 compared to empty vector using one-way ANOVA).

**Figure 3 ijms-21-03047-f003:**
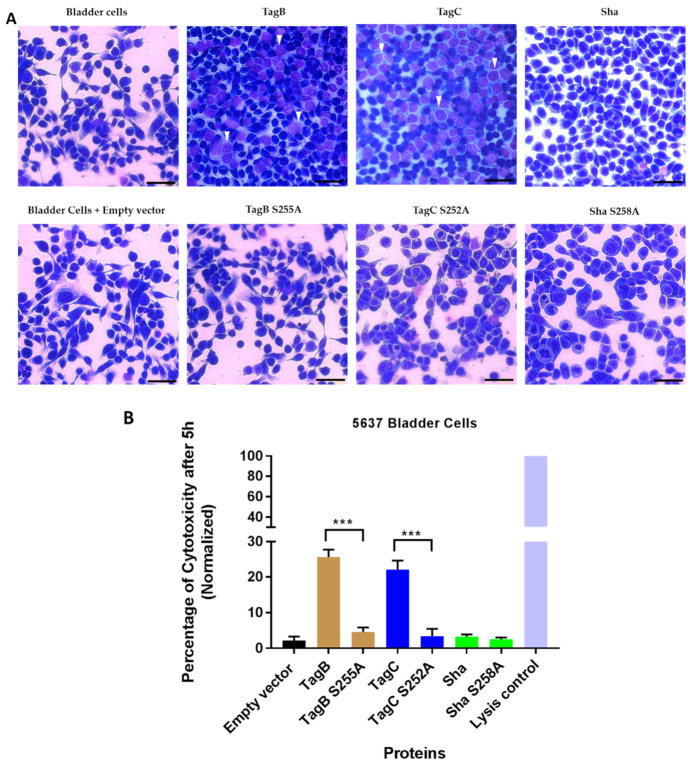
The serine catalytic site is necessary for the cytopathic effect of TagB and TagC. (**A**) Concentrated supernatants containing 30 μg of protein per well derived from *E. coli* BL21 clones expressing TagB, TagC, Sha, or their respective serine mutant variant proteins were incubated with monolayers of the 5637 human bladder epithelial cell line for 5 h at 37 °C. Cytopathic effects (white triangle) were absent in cells treated with the serine catalytic site mutant variants of TagB or TagC. The empty vector (pBCsk+) without insert was used as a negative control. The scale bar represents 20 µm. (**B**) Cytotoxicity measured by LDH release from 5637 human bladder cells after incubation with supernatant filtrates of different clones (30 μg of protein per well) at 37 °C for 5 h. Empty vector (pBCsk+) was used as a negative control and maximum LDH release (positive control) was determined by treatment with lysis solution. Data are the means of three independent experiments, and error bars represent the standard errors of the means. Significant differences between lysis caused by native and mutant SPATEs were determined using Student’s *t*-test with *** *p* < 0.001.

**Figure 4 ijms-21-03047-f004:**
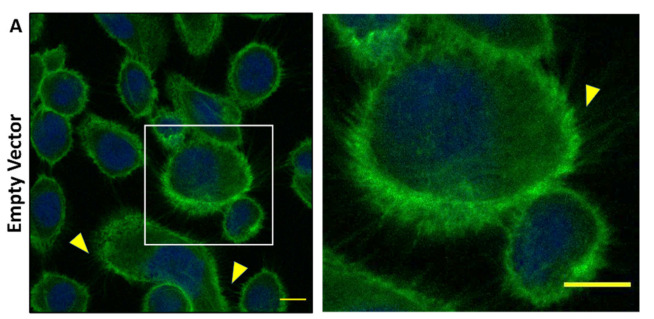
Effects of TagB, TagC, and Sha on the actin cytoskeleton of bladder epithelial cells is serine-protease-motif dependent. (**A**) Concentrated supernatant extracts (30 μg of protein per well) from *E. coli* BL21 clones expressing TagB, TagC, or Sha and their respective serine catalytic site mutants were incubated with monolayers of human bladder (5637) epithelial cells for 5 h at 37 °C. After incubation, cells were fixed and permeabilized. Actin was stained with fluorescently labeled phalloidin (green) and the nucleus was stained by DAPI (blue). Cells treated with the filtered supernatant of *E. coli* BL21 pBCsk+ without insert (empty vector) were used as a negative control. Slides were observed by confocal microscopy. Inset images from the left panels are magnified in the panels to the right. Bars represent 10 µm. (**B**) Quantitative analysis of fluorescence intensity of F-actin. Analysis of fluorescent intensity was done at the original magnification by measuring the mean gray value with ImageJ software [22] with an *n* value of at least 10 cells. Data values represent the mean ± SEM of at least three independent experiments. (* *p* < 0.05, ** *p* < 0.01, *** *p* < 0.001 one-way ANOVA with multiple comparisons).

**Figure 5 ijms-21-03047-f005:**
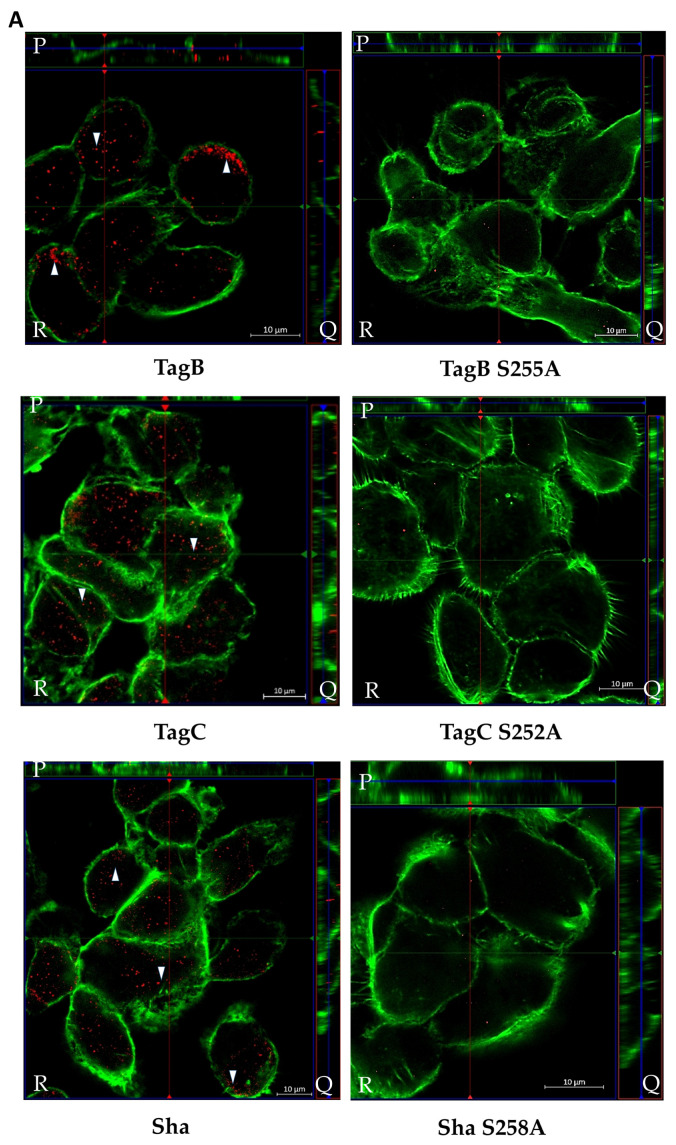
Intracellular localization of TagB, TagC, and Sha determined by confocal microscopy. (**A**) Z-stack imaging showing the localization of TagB, TagC, and Sha and their respective serine active site mutant variants during interaction with 5637 bladder epithelial cells after 5 h of incubation. SPATEs were detected by Alexa Fluor 594 (white arrowheads, red fluorescence) using anti-mouse secondary antibody and actin was stained by Alexa Fluor 488- phalloidin (green fluorescence). Images are displayed in a 3D section view with large Z-sections in the X-Y direction (R), Z-projection in the X–Z direction (P), and Z-projection in the Y–Z direction (Q). The green and red lines in R indicate the orthogonal planes of the X–Z and Y–Z projection. For each selected section, the signal was gathered from a span of 5 μm. Scale bar: 10 µm (**B**) Quantitative analysis of fluorescence intensity of F-actin in the cells treated with native or mutant SPATEs. Analysis of fluorescence intensity was done in green channel by measuring the mean gray value on ImageJ. Data represent the mean ± SEM of at least three independent experiments. Significant differences between fluorescence intensity of each native and mutant SPATE treated cell was determined using Student’s t-test with ** *p* < 0.01, *** *p* < 0.001.

**Figure 6 ijms-21-03047-f006:**
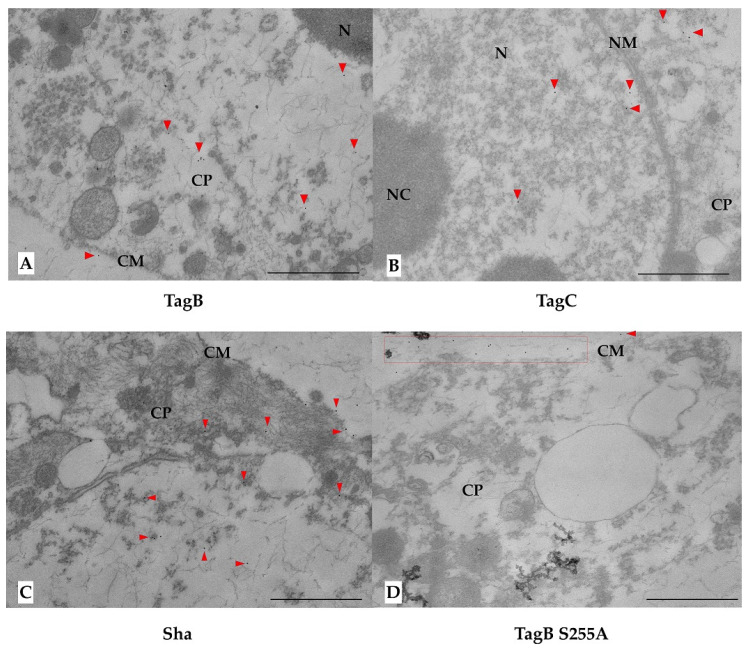
Transmission electron micrographs of 5637 bladder cells showing internalized SPATEs, immunolabelled with 10-nm-diameter gold particles after 5 h of incubation. Gold particles are highlighted with red triangles. (**A**) TagB is principally located in the cytoplasm (CP). (**B**) For Tag C, gold particles were associated with the nucleus (N) and cytoplasm (CP). (**C**) Sha was located mainly in the cytoplasm (CP). (**D**) For the serine mutant variants of TagB, TagC, and Sha, gold particles were only localized on the extracellular surface of cells (red box). Only the TagB S255A mutant protein localization is shown. Cell membrane (CM), Cytoplasm (CP), Nuclear Membrane (NM), Nucleus (N), Nucleolus (NC) Bars, 1 µm.

**Figure 7 ijms-21-03047-f007:**
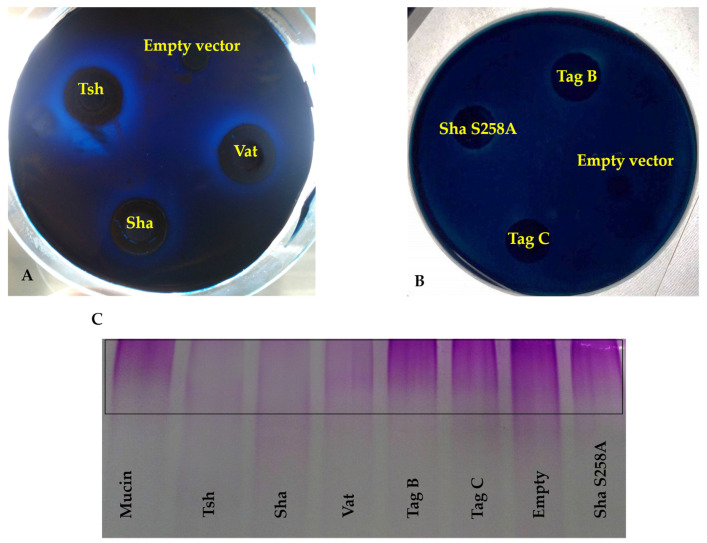
Sha demonstrates serine-protease dependent mucinase activity, but not TagB nor TagC. Mucinase activity was tested in a medium containing 1.5% agarose and 0.5% porcine gastric mucin. Filter discs inoculated with clones containing the empty vector, expressing Sha, Vat, Tsh, (**A**) TagB, TagC, or Sha S258A (**B**) were placed on the agar surface and incubated overnight at 37 °C. Mucin lysis zones were visualized by staining with 0.1% amido-black in 3.5 M acetic acid for 15 min, followed by destaining with 5% acetic acid and 0.5% glycerol for 6 h to overnight. (**C**) Zones of 0.5% porcine gastric mucin hydrolysis are visible in the stacking region of the SDS-PAGE gel (boxed area), concentrated supernatant extracts of SPATEs (5 μg of protein per well) were incubated at 37 °C for 48 h with mucin prior to migration. The gel was stained with a PAS glycoprotein staining kit.

**Figure 8 ijms-21-03047-f008:**
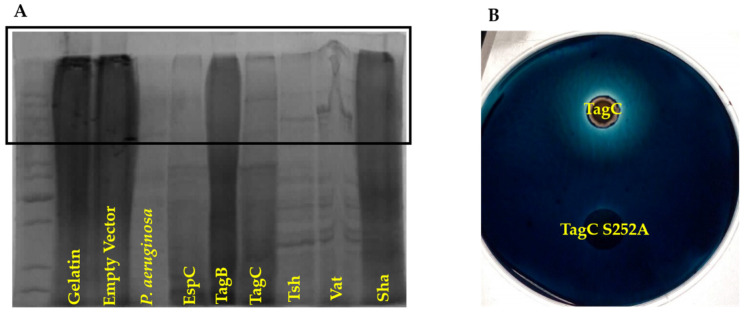
TagC demonstrates serine-protease dependent gelatinase activity, but not TagB nor Sha. (**A**) Zones of 1% bovine skin gelatin hydrolysis are visible in the stacking region of the SDS-PAGE gel (boxed area), concentrated supernatant extracts of SPATEs (5 μg of protein per well) were incubated at 37 °C for 48 h prior to migration. (**B**) Gelatinase activity of TagC was tested in a medium containing 1.5% agarose and 1% bovine skin gelatin. The disc inoculated with a clone expressing TagC or its serine catalytic site mutant variant, TagC S252A, were inoculated on the agar surface and were incubated for 48 h at 37 °C. Zones of gelatin lysis were visualized by staining with 0.1% amido-black in 3.5 M acetic acid for 15 min, followed by destaining with 5% acetic acid and 0.5% glycerol for 6 h.

**Table 1 ijms-21-03047-t001:** Strains and plasmids used in this study.

Strains	Characteristic(s)	Reference
QT598	APEC O1: K1, serum resistant	[61]
ORN172	*thr-1 leuB thi-1*Δ *(argF-lac)U169 xyl-7 ara-13 mtl-2 gal-6 rpsL tonA2 supE44*Δ *(fimBEACDFGH)::Km pilG1*	[62]
BL21	*fhuA2 [lon] ompT gal [dcm] ΔhsdS*	New England Biolabs
QT1603	HB101 with AIDA-1 operon	[63]
QT4767	ORN172/pIJ553 (Expressing *sha*)	[20]
QT5194	BL21/pIJ548 (Expressing *tagB*)	
QT5195	BL21/pIJ549 (Expressing *tagC*)	
QT5197	BL21/pIJ550 (Expressing *espC)*	
QT5198	ORN172/pIJ548 (Expressing *tagB*)	
QT5199	ORN172/pIJ549 (Expressing *tagC*)	
QT5431	BL21/pIJ551 (Expressing *vat*)	
QT5432	BL21/pIJ552 (Expressing *tsh*)	
QT5433	BL21/pIJ553 (Expressing *sha*)	
QT5437	BL21 + pIJ554 (Expressing *tagB S255A*)	This study
QT5438	BL21 + pIJ555 (Expressing *tagC S252A)*	This study
QT5439	BL21 + pIJ556 (Expressing *sha S258A)*	This study
QT5598	ORN172 + pIJ554 (Expressing *tagB S255A*)	This study
QT5599	ORN172 + pIJ555 (Expressing *tagC S252A)*	This study
QT5600	ORN172 + pIJ556 (Expressing *sha S258A)*	This study
QT3046	*Pseudomonas aeruginosa* PA14	Eric Déziel, INRS
Plasmids		
pBCsk+	Cloning vector; Cm^r^	Stratagene, La Jolla, CA
pIJ548	pBCsk+::*tagB*	[20]
pIJ549	pBCsk+::*tagC*	
pIJ551	pBCsk+::*vat*	
pIJ552	pBCsk+::*tsh*	
pIJ553	pBCsk+::*sha*	
pIJ554	pBCsk+::*tagB S255A*	This study
pIJ555	pBCsk+::*tagC S252A*	This study
pIJ556	pBCsk+::*sha S258A*	This study

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
