# Peer review of "The Serine Protease Autotransporters TagB, TagC, and Sha from Extraintestinal Pathogenic Escherichia coli Are Internalized by Human Bladder Epithelial Cells and Cause Actin Cytoskeletal Disruption"

_ijms, 2020, doi:10.3390/ijms21093047_

Round 1
Reviewer 1 Report
This paper examines a number of phenotypes associated with three newly identified SPATEs, TagB, TagC, and Sha. All three toxins promoted bacterial cell autoaggregation, independent of their serine protease activities. Each toxin also caused cytopathic effects in cultured bladder cells, with possible effects on the actin cytoskeleton. Finally, the authors show that TagC can degrade gelatin while Sha can break down mucin. Most of the results are well substantiated and shed light on the functions of these toxins, though the phenotypes reported are not unexpected based on work with other SPATEs. There are some additional controls and changes to the text and figures that should be included to back up key conclusions, as described below.
- The data could be presented much more clearly and concisely. The text has numerous grammatical errors and is unnecessarily convoluted. Often, this made it difficult to follow what the authors are trying to convey. Some re-writing of the text would make this study much more accessible to a broader audience.
- For Figure 1, insets showing close-up of the immunogold-labeling would be helpful, It is hard to appreciate the distribution of the particles as presented. It is not clear why the secreted passenger domains are so concentrated at the bacterial surface. The labeling in C and D suggests that TagC and Sha are associated with large fiber-like structures. Are the SPATEs aggregating around the bacteria? Are the SPATE passenger domains inefficiently cleaved from the bladder cells? Does this contribute to the autoaggregation effects (figure 2)?
- Scale bars in Figures 1 and 3 are too small to appreciate. Furthermore, they do not appear to be accurate.
- For the top panel of Figure 3, the differences in confluency between control and toxin-treated samples makes it difficult to make comparisons. The data would be more convincing if the cytopathic effects were quantified
- The authors’ conclusion that the SPATEs disrupt the actin cytoskeleton, based on representative images presented in Figure 4, requires further substantiation. It is not clear if the cells were all imaged at the same level of focus. Quantitative analysis of phalloidin binding to F-actin using a fluorometer or plate reader would greatly strengthen these findings.
- The authors use immunofluorescence and TEM with immunogold labeling in figures 5 and 6 to make the case that the SPATEs are internalized into host cells. However, at the 5-hour time point examined, there is a lot of cytotoxicity (based on figure 3). How do the authors know that the toxins are not just entering damaged host cells with compromised membranes? The microscopy is not especially convincing. Again, quantification of results using blinded analyses of samples would strengthen the conclusions. Based on the TEM, Sha is present within both nuclei and the cytoplasm, though the text makes it seem as though it is all nuclear. The immunofluorescent imaging does not support the idea that Sha localizes mostly within the nuclei. Imaging and quantifying Sha localization with a nucelar dye would be helpful.
- The authors treat bladder cells with 30 ug of the toxins. Can the authors comment on whether or not this physiologically relevant?
- The authors conclusion that Sh has a mucin-degrading phenotype intermediate between Tsh and Vat is supported by the gel shown in Fig 7C, but not by the plat in figure 7A. Why is the gel in 7c so crooked?
- I n Figure 8A, use of P. aeruginosa comes out of nowhere. Why is this used as a positive control in this assay?
- In a couple of places the authors refer to conserved epitopes in the passenger domains of SPATEs. This should be referenced or better substantiated.
- Table 1 is not needed.
- Line 266-268, as written, suggest that mucin treated with the mutant SPATE prevents the wild type Sha toxin from degrading mucin (as if the mutant Sha prevent wild type Sha from acting on mucin). This would be interesting, but is not supported by the data and is not what I think the authors meant to imply. Please clarify.
- The first paragraph of the Discussion covers a lot material that is already presented in the Introduction. It is not needed.
- Lines 332-337 in the Discussion are very confusing, and seems to contain incorrect information.
- Line 134, AIDA-1 should be defined.
- Lines 180-81, should be “To examine the disruption…”.
Lines 188-89. “…cytoplasm was detached from the substratum” This does not make sense.
Line 281, needs a reference.
Line 420, needs a reference.
Reviewer 2 Report
Minor revision:
Please, rephrase sentence in lines 87-90.
Is there any possibility to hypothesize the mechanism by which these SPATEs are released in the extracellular milieu independently from their serine-protease activity? I would like to suggest to perform a bioinformatic analysis by aligning different SPATEs sequences and possibly identifying hypothetical amino acid residues needed for their cleavage.
Line 99, it is not clear why authors raised antibody against Vat to localize SPATEs Tag B/C and Sha. Can the authors explain this point?
Figure 1: bars are not visible, can you convert them in 1 µm size to make them readable? The image dimensions are different, is there any possibility to modify them using the same size? In hìthe figure legend title E. coli should be written in italic.
The results showed that Tag B/C and Sha S to A mutants autoaggregate and agglutinate at the same extent of wild type proteins, I would like to know if the Tag B/C and Sha catalytic activity is linked to the capability of bacteria to adhere to eukaryotic cells.
Major revision:
Figure 3: the evaluation of cytophatic effect is a little bit difficult due to the diverse cell concentration among the different conditions tested. Since the authors previously published the LDH release assay to quantify the cytotoxicity of these proteins, can they perform the same experiment? Vice versa, is there any possibility to quantify this cytotoxic effect using for example the MTT test or a simple cell count with trypan blue? The quantification of toxic effect could clarify the qualitative analysis of cell morphology.
The cytophatic effect of these proteins is probably linked to their activity on cytoskeletal actin however, from the data presented it is not clear which kind of effect they should exert on actin. Can the authors explain how they evaluated the “reduced actin stress fibers and the less actin staining around the cell” due to the SPATEs action?
Figure 4: The evaluation of cytoskeletal alteration is difficult just looking at the proposed images, probably an enlargement of each field could help in visualizing the altered phenotype. Since the authors acquired images using the confocal microscope I suggest to acquire images with higher magnifications.
I can also suggest to visualize this phenotype using another cell line in which the stress fibers and cortical actin are more easily detectable, for example HeLa cells are widely used for cytoskeletal studies. Moreover, adding a Cytochalasin-treated cells, representing a positive control of actin disruption, would be helpful for result evaluation.
The authors elegantly demonstrated that the intracellular localization of TagB/C and Sha requires the catalytic activity. However, their activity on cytoskeletal actin is not clear looking at the images proposed (Figure 5).
Minor revision:
Lines 338-359: can be greatly shorten because the authors demonstrated that autoaggregation and agglutination phenotypes did not depend on the SPATE catalytic activity however, they did not provide any hypothesis about the biological function of these proteins.
Lines 398-391: since the catalytic activity of these SPATEs targets host actin the authors concluded that they are not involved in bacterial cell invasion. However, in the previously published paper they demonstrated the increased SPATEs-mediated bacterial adherence to cells. Moreover, they observed an early cytopathic effect of TagB/C that decreases over time. Hence, it can be also hypothesized that this activity helps the bacterial cell invasion acting on actin remodeling known to be required for UPEC invasion. Can the authors add a comment about this point?
Finally, the authors demonstrated the proteolytic activity of these SPATEs on component of the extracellular matrix such as mucin and gelatin highlighting this conserved activity among SPATE members. It could be highly interesting to evaluate the double role of this proteins according to their different localization. They can help the bacterial colonization of tissue when they are released extracellularly: vice versa they can act on cell components when trafficking intracellularly. Can the authors add comments about this point in the discussion?
Overall the discussion section is too long and should be shorten.
Why the authors used a starting concentration of bacterial cell corresponding to an optical density of 60 for the hemagglutination assays? Did they follow some published protocol?
Round 2
Reviewer 2 Report
Overall the points raised were properly addressed. There are some typing mistakes and minor errors:
line 80: delete "in"
line 165: Concentrated supernatants containing 30 g of protein per well derived from E. coli BL21 clones
lines 250-251: Significant differences between lysis caused by native and mutant SPATEs were determined.....; this sentence has no sense, it is a mistake.
Author Response
Already revised accordingly